# Sponge Communities of Submarine Caves and Tunnels on the Fernando de Noronha Archipelago, Northeast Brazil

Guilherme Muricy [1,2,*], Anaíra Lage [1,3], Joana Sandes [1], Michelle Klautau [3], Ulisses Pinheiro [4], Marinella Silva Laport [5], Bruno Francesco Rodrigues de Oliveira [6], Carolline Braga Pequeno [1] and Matheus Vieira Lopes [3]

1    Museu Nacional, Universidade Federal do Rio de Janeiro, Rio de Janeiro 20940-040, RJ, Brazil; anairalage@gmail.com (A.L.); jcfsandes@gmail.com (J.S.); carolline.pequeno@gmail.com (C.B.P.)
2    Instituto de Pesquisas Jardim Botânico do Rio de Janeiro, Rio de Janeiro 22460-030, RJ, Brazil
3    Laboratório TaxoN, Departamento de Zoologia, Universidade Federal do Rio de Janeiro, Rio de Janeiro 21941-599, RJ, Brazil; mklautau@gmail.com (M.K.); mvlopes@ufrj.br (M.V.L.)
4    Departamento de Zoologia, Universidade Federal de Pernambuco, Recife 50373-970, PE, Brazil; ulisses.pinheiro@ufpe.br
5    Instituto de Microbiologia Paulo de Góes, Universidade Federal do Rio de Janeiro, Rio de Janeiro 21941-902, RJ, Brazil; marinella@micro.ufrj.br
6    Departamento de Microbiologia e Parasitologia, Instituto Biomédico, Universidade Federal Fluminense, Niterói 24210-130, RJ, Brazil; bfroliveira@id.uff.br
*    Correspondence: muricy@mn.ufrj.br

**Abstract:** Submarine caves are important biodiversity reservoirs, but there is little information about the biota of marine caves in the Southwestern Atlantic. Here, we describe three submarine cavities and their sponge communities on the Fernando de Noronha Archipelago, Northeast Brazil. The underwater cavities were explored and collections were made through scuba diving from 5 to 18 m depths. Sapata Cave has a wide semi-dark zone near the entrance, a narrow transition zone, and a dark chimney, which is closed at the top. Ilha do Meio Cave is narrower and shallower than Sapata Cave, but has a long passage that leads to two completely dark rooms. Pedras Secas Tunnel has only a semi-dark zone with high water movement. The sponge communities in the semi-dark zones of the three cavities are rich and dominated by the classes Demospongiae and Homoscleromorpha, but Calcarea are also common. The transition zones of both caves are dominated by a desma-bearing sponge, thinly encrusting spirastrellids, and small Homoscleromopha and Calcarea. The dark zone in Ilha do Meio Cave is almost azoic, with only three species. This study has increased the number of sponge species known in submarine cavities on Fernando de Noronha from 29 to 69, highlighting the great richness of the sponge communities in these cryptic environments.

**Keywords:** Tropical Western Atlantic; oceanic island; Brazil; submarine caves; tunnels; Porifera; diversity; zonation

## 1. Introduction

Caves are special environments of great ecological, biogeographic, paleontological, geological, archaeological, and touristic importance [1–3]. Due to the great limitation of light, there are no plants to feed the herbivores. Therefore, there is little food available, and only very specialized animals can live in caves [4]. Cave ecosystems are fragile, but provide efficient shelter for the conservation of endemic and endangered species [5]. Despite their importance, terrestrial and freshwater caves are still little studied, and even less is known about the ecosystems of marine and anchialine caves [6,7].

Underwater caves can only be explored through scuba diving, and, therefore, their study has only been possible since the 1950s, after the invention of diving cylinders or "aqualungs" [6,8]. Thanks to modern scientific diving methods, submarine caves have been discovered to be hotspots and reservoirs of biodiversity [9].

Submarine caves, grottoes, and tunnels are peculiar ecosystems due to their similarities with abyssal plains [10,11]. Their limited light, oligotrophy, low water circulation, and, in some cases, cold and constant temperature allow the colonization of shallow water caves by organisms normally found only in the deep sea. Therefore, marine caves act as refuges and allow *in situ* studies of rare and difficult-to-access species from the deep sea [12,13].

Not many organisms can adapt to survive in the environments of submarine caves. In general, the most common benthic animals in marine caves are sponges (Porifera). Due to their sessility and filtering habit, sponges can survive by feeding only on the few planktonic microorganisms available in caves. Thus, underwater caves often harbor abundant and diverse sponge communities [9,14,15]. In general, sponge species from submarine caves are different from those inhabiting adjacent rocky shores. For example, many species of the classes Homoscleromorpha and Calcarea are often more abundant in caves and other cryptic environments than on illuminated reefs and rocky shores [6,16]. Furthermore, cave sponges often comprise taxa that are rare nowadays, but were abundant in past eras [17,18]. These relics of prehistoric fauna, including sponges with desmas (formerly "Order Lithistida") and sponges with a hypercalcified skeleton such as Sphinctozoa, Pharetronida, and Chaetetida [16,19–21], are of great interest to taxonomists and paleontologists. Thus, studies on cave-dwelling sponges can greatly expand the current knowledge about the diversity, biology, and evolution of animals.

Most studies on cave sponges have been carried out in the Mediterranean Sea, the Tropical Northwestern Atlantic (especially Bermuda, the Bahamas, and the Yucatan Peninsula in Mexico), and in the Pacific Ocean, where submarine caves are very common [6,7,9,15–17,22–25]. The diversity of cave-dwelling species worldwide is currently being compiled in the WoRCS database (World Register of marine Cave Species [26]). In Brazil, most studies on underwater caves focus on freshwater caves, which are quite common in the country [27,28]. In contrast, little is known about submarine caves and their biota in Brazil, whose large littoral encompasses most of the Tropical Southwestern Atlantic coasts. Only five localities in Brazil are known to have submarine caves, and all of them are still poorly studied. Off Rio de Janeiro city, there is a semi-submerged cave called "Buff", located on Filhote da Redonda Island [29]. Although underwater photographs show a rich benthic fauna inside, there are no published descriptions of its benthic communities so far. In Arraial do Cabo, a semi-submerged cave named "Gruta Azul" is a famous touristic attraction. Fifteen sponge species were reported from this cave, mostly of the class Calcarea and a few Demospongiae [30–32]. There are other submarine caves in Arraial do Cabo, such as "Gruta da Camarinha" and "Buraco dos Meros", but their benthic communities remain unexplored. On the Abrolhos Archipelago, there is a large submerged cave on Siriba Island, from which a single sponge species has been reported so far. Furthermore, on the Rocas Atoll, there are two large tunnels (named "Fendas 1 and 2") colonized by a rich sponge community, with 17 sponge species reported to date. Finally, on the Fernando de Noronha Archipelago there are many submarine caves and tunnels, which are still poorly known.

Up to this study, only 29 sponge species were known to inhabit the submarine cavities of Fernando de Noronha. All records came from two submarine caves, namely "Sapata Cave" (8 spp.) and "Ilha do Meio Cave" (4 spp.), and one tunnel, "Pedras Secas Tunnel" (20 spp.; [33–39]). However, there were no published descriptions of the geomorphologies of these caves and tunnels or of their biotic communities.

Therefore, the goals of this work were to describe the main submarine caves and tunnels of Fernando de Noronha and to characterize their sponge communities in terms of species composition, richness, and zonation. This is the first systematic report unveiling the sponge fauna in submarine caves of this archipelago and in the whole Tropical Southwestern Atlantic region.

## 2. Materials and Methods

### 2.1. Study Area

Fernando de Noronha, nicknamed "the Emerald of the Atlantic", is an isolated volcanic archipelago formed by a magmatic hotspot 12 Mya [40]. It is located at 03°51′ S–32°25′ W, 360 km NE of Natal city, in NE Brazil (Figure 1A). The climate is tropical, with the air temperature ranging from 25 to 30 °C and water temperature varying from 24 to 30 °C. It has a rich marine biota, protected by the National Marine Park of Fernando de Noronha [41]. The archipelago occupies an area of 26 km$^2$ and is composed of a main island, called Fernando de Noronha, and 20 islets (www.noronha.pe.gov.br; accessed on 6 March 2024). The main island is 10 km long, with a maximum width of 3.5 km. The archipelago has many sandy beaches, rocky coasts, and tidal pools and several submarine caves and tunnels [42]. The sea on the northwestern side of Fernando the Noronha is called "Mar de Dentro" (Portuguese for "Inside Sea"; Figure 1B–D). It is relatively calm and sheltered during the autumn and winter months (from April to September), but is usually subjected to strong swells in spring and summer (from October to March). The sea on the southeastern side of the island is called "Mar de Fora" (Portuguese for "Outside Sea"; Figure 1B–D) and is more exposed to strong waves and currents during the whole year.

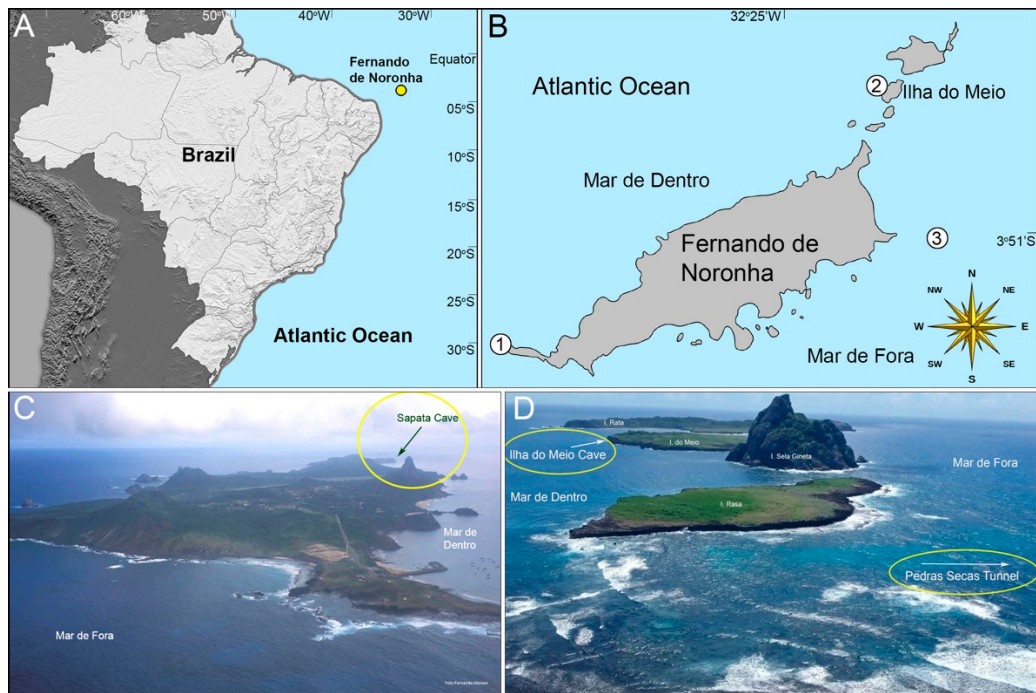

**Figure 1.** Study area. (**A**) Location of Fernando de Noronha Archipelago. (**B**) Location of the collection sites: 1, Sapata Cave. 2, Ilha do Meio Cave. 3, Pedras Secas Tunnel. (**C**) Aerial view of the main island of Fernando de Noronha with its two sides, the more exposed "Mar de Fora" ("Outer Sea" in Portuguese) facing southeast and the more sheltered "Mar de Dentro" ("Inner Sea" in Portuguese) facing northwest, and highlighting the location of Sapata Cave in the southwest tip of the island (arrow). (**D**) Aerial view of the islets on the NE side of the archipelago, highlighting the location of Ilha do Meio Cave and the direction to Pedras Secas Tunnel (arrows).

### 2.2. Methods

Two expeditions to the Fernando de Noronha Archipelago were organized in May and November 2022. Collections were made through scuba diving from 3 to 20 m depths in three submarine cavities (Figure 1B–D): Sapata Cave (3.874921 °S, 32.476105 °W), Ilha do Meio Cave (3.818404 °S, 32.393399 °W), and Pedras Secas Tunnel (3.851128 °S, 32.381342 °W). Most sponges collected were cut in three parts: fixed in 70% ethanol for morphology, 2% glutaraldehyde in 0.4 M sodium cacodylate buffer for cytology (on board, soon after

collection), and CHAOS solution for DNA extraction (4-M guanidine thiocyanate, 0.5% *N*-lauryl sarcosil, 25-mM Tris pH 8.0, 01-M β-mercaptoethanol).

The species were identified through morphological analysis. Full descriptions including morphological, cytological, and molecular characteristics will be published elsewhere. Species reported in previous studies [33,37–39] were considered in the descriptions of these sponge communities. The specimens collected were deposited at the Porifera collections of the Museu Nacional (MNRJ) and Instituto de Biologia (UFRJPOR), both of the Universidade Federal do Rio de Janeiro, and the Porifera collection of the Universidade Federal de Pernambuco (UFPEPOR).

## 3. Results

### *3.1. Sapata Cave*

#### 3.1.1. Description of the Cave

Sapata Cave is located on the SW tip of the main island of Fernando de Noronha (Figure 1B,C), at the end of a promontory made of pyroclastic rocks with vertical walls up to 80 m high [43]. Underwater, these vertical walls continue down to 10–15 m depths, with large boulders deposited from 15 to 30 m depths in front of the cave [44]. The cave entrance is relatively large, approximately 10 m high by 20 m wide, ranging from 7 to 17 m depths. The bottom is sandy and nearly horizontal. The cave can be divided in three sections (Figures 2 and 3): (1) a semi-dark zone near the entrance, approximately 20 m wide by 15 m long and 10 m high; (2) a transition zone, at a distance of 15–30 m from the entrance, with low light; and (3) a roughly conical, vertical chimney, approximately 3 m in diameter by 8 m high, in the ceiling of the transition zone. The chimney could not be explored in this study, but it probably represents a true dark zone. The cave has a single entrance and, therefore, the water circulation inside it is very weak.

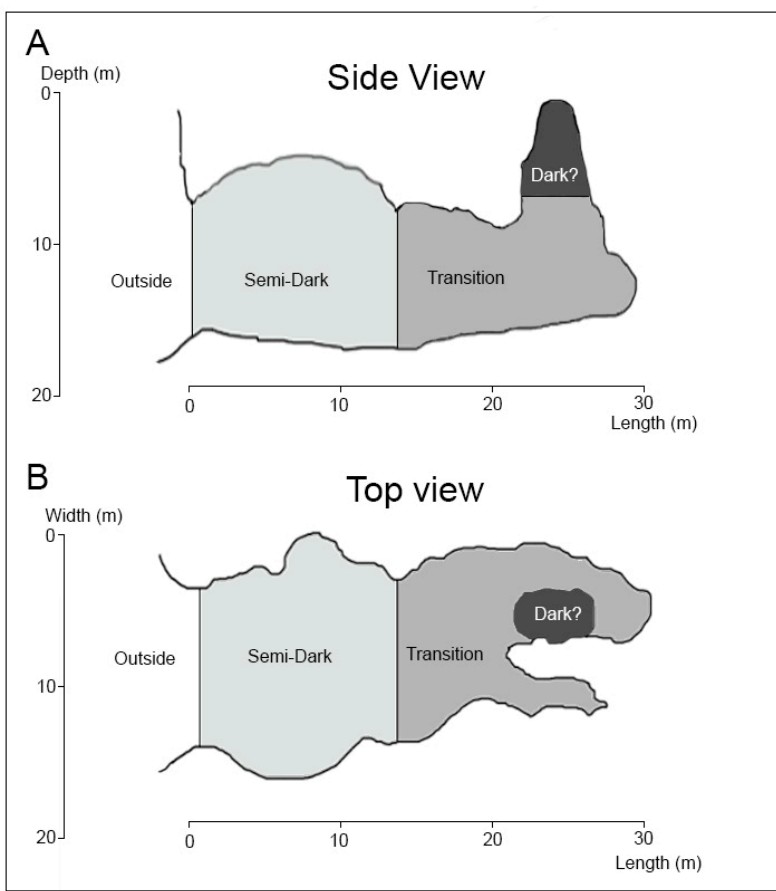

**Figure 2.** Schematic sections of Sapata Cave. (**A**) Lateral view. (**B**) Top view.

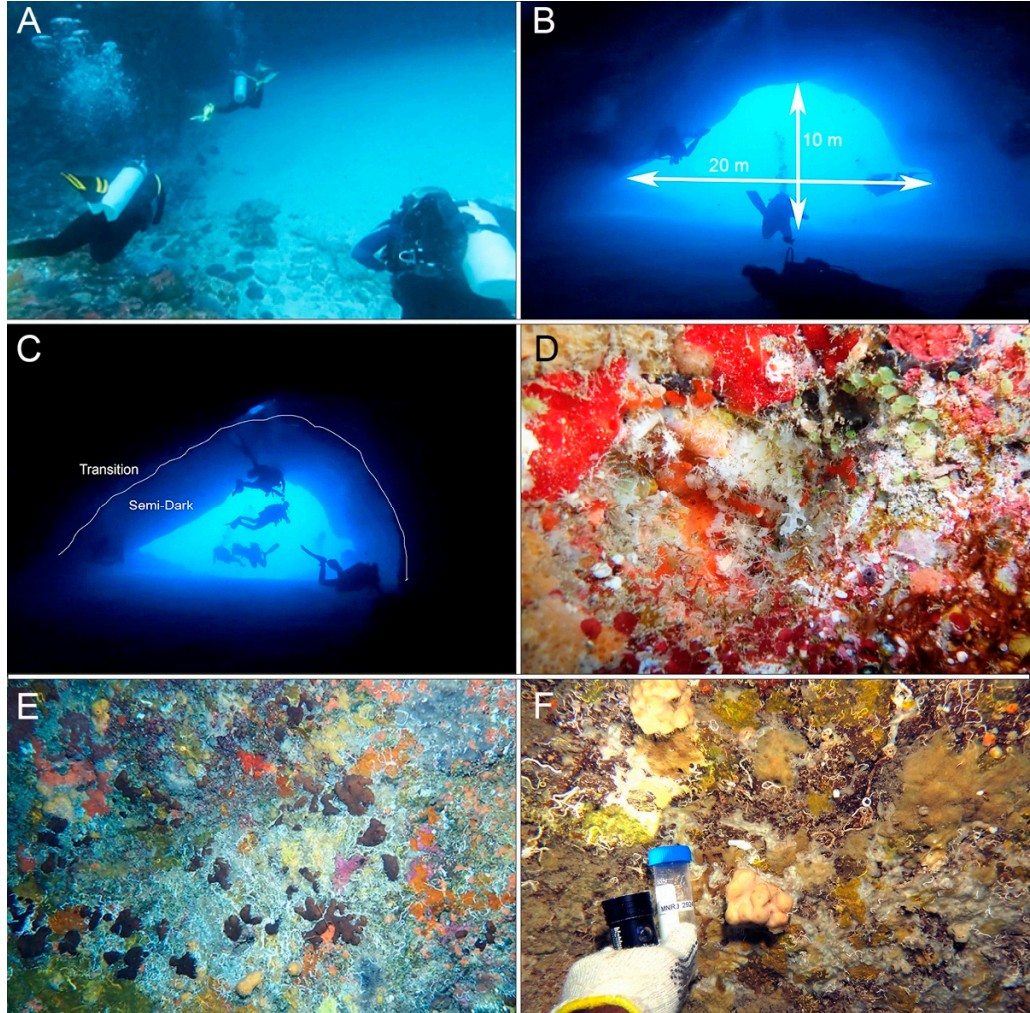

**Figure 3.** Sapata Cave. (**A**) Sandy bottom at the entrance of the cave. (**B**) Entrance of the cave viewed from inside. (**C**) Light gradient between semi-dark and transition zones. (**D**) Close-up of the benthic community inside a small crevice outside the cave. (**E**) Wide-angle view of the benthic community in the semi-dark zone, dominated by the sponges *Plakortis insularis* (brown) and *Spirastrella hartmani* (orange). (**F**) Benthic community in the transition zone, dominated by the massive "lithistid" *Gastrophanella cavernicola* and thinly encrusting demosponges (*Diplastrella megastellata*, *Jaspis* sp.).

Sapata Cave is a shelter for large fish such as the Southern Stingray *Hypanus americanus* [45] and the Atlantic Goliath *Grouper Epinephelus itajara* [46].

### 3.1.2. Sponge Communities of Sapata Cave

Outside Sapata Cave, the sponge community is rich and colorful. We found 13 species, but there are probably many more. Most species outside the cave are common reef sponges, widely distributed in the Tropical Western Atlantic Ocean, such as *Plakortis angulospiculatus*, *Monanchora arbuscula*, *Ectyoplasia ferox*, *Ircinia strobilina*, *I. felix*, and *Agelas dispar* (Figure 4A–D, Table 1). Calcareous sponges were also diverse and abundant in crevices outside the cave (Figure 3D).

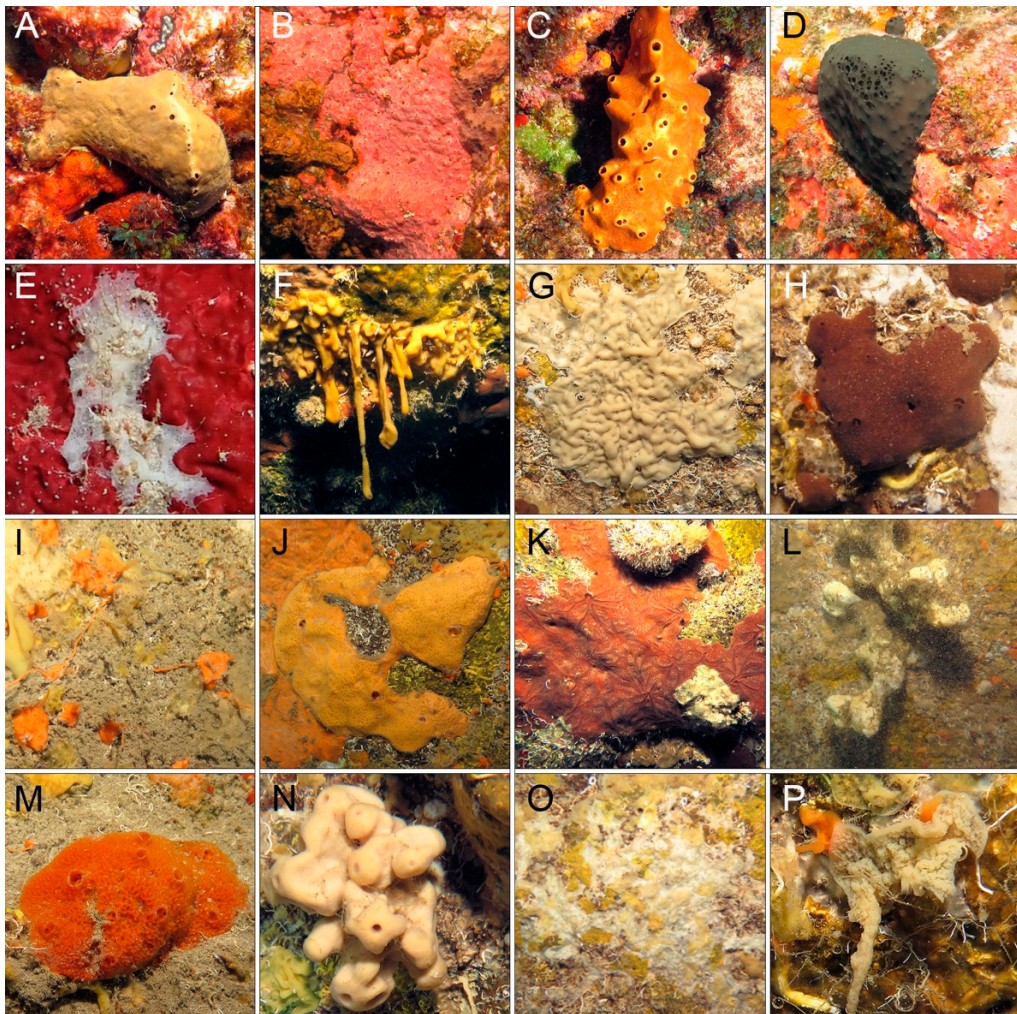

**Figure 4.** *In situ* photographs of representative sponge species living inside and outside Sapata Cave. (**A–D**) Outside the cave: (**A**) *Plakortis microrhabdifera*. (**B**) *Monanchora arbuscula*. (**C**) *Ectyoplasia ferox*. (**D**) *Ircinia strobilina*. (**E–M**) Semi-dark zone: (**E**) *Clathrina delicata*. (**F**) *Oscarella* cf. *zoranja*. (**G**) *Plakina* sp. (**H**) *Plakortis insularis*. (**I**) *Oceanapia* sp. (**J**) *Ectyoplasia ferox*. (**K**) *Spirastrella hartmani*. (**L**) *Agelas* sp. (**M**) *Dragmacidon reticulatum*. (**N–P**) Transition zone: (**N**) *Gastrophanella cavernicola*. (**O**) *Jaspis* sp. (white) and *Diplastrella megastellata* (yellow). (**P**) *Oscarella* sp.

**Table 1.** Occurrence and semi-quantitative abundance of sponge species in each zone inside and outside three submarine cavities studied on Fernando de Noronha (including previously published records and new findings). Abundance: xxx, abundant; xx, common; x, rare.

| | Sapata Cave | | | Ilha do Meio Cave | | | | Pedras Secas Tunnel | |
|---|---|---|---|---|---|---|---|---|---|
| **Species** | **Outside** | **Semi-Dark Zone** | **Transition Zone** | **Outside** | **Semi-Dark Zone** | **Transition Zone** | **Dark Zone** | **Outside** | **Semi-Dark Zone** |
| CLASS DEMOSPONGIAE | | | | | | | | | |
| *Agelas clathrodes* [47] | x | | | | | | | xx | xxx |
| *Agelas dispar* [48] | xx | | | xxx | x | | | xx | xxx |
| *Agelas* sp. 1 | | x | | | | | | | |
| *Agelas* sp. 2 | | | | | x | | | | |
| *Aiolochroia crassa* [49] | | | | xx | | | | xx | x |
| *Amphimedon compressa* [48] | | | | xx | x | | | x | |
| *Aplysina fulva* [50] | | | | | | | | x | |
| *Batzella* sp. 1 | | x | | | | | | | x |
| *Batzella* sp. 2 | | x | | | | | | | |

**Table 1.** *Cont.*

| Species | Sapata Cave | | | Ilha do Meio Cave | | | | Pedras Secas Tunnel | |
| --- | --- | --- | --- | --- | --- | --- | --- | --- | --- |
| | Outside | Semi-Dark Zone | Transition Zone | Outside | Semi-Dark Zone | Transition Zone | Dark Zone | Outside | Semi-Dark Zone |
| *Chondrilla caribensis* [51] | | | | | | | | | x |
| *Chondrosia collectrix* [47] | | | | | | | | | x |
| *Chondrosia* sp. | | | | x | | | | | |
| *Clathria (Thalysias) minuta* [52] | | | | | | | | xx | x |
| *Dercitus (Halinastra) luteus* [53] | | | | | | | | | x |
| *Dercitus (Stoeba) latex* [36] | | | | | x | | | | |
| *Dictyonella* sp. | | | | | | | | | x |
| *Diplastrella megastellata* [54] | | xxx | xx | | | xx | x | | |
| *Dragmacidon reticulatum* [55] | | x | | | | | | | |
| *Dysidea* sp. | | x | | | | | | | |
| *Ectyoplasia ferox* [48] | xx | xx | | | | | | | |
| *Erylus formosus* [56] | | | | | | | | | x |
| *Gastrophanella cavernicola* [34] | | | xx | | | xx | | | |
| *Haliclona* sp. | | | | | x | | | | |
| *Halisarca caerulea* [57] | | | | x | | | | | |
| *Hyrtios proteus* [48] | | | | | | | | | x |
| *Ircinia felix* [48] | x | | | x | | | | xx | |
| *Ircinia strobilina* [58] | xx | | | xxx | | | | xx | x |
| *Ircinia* sp. 1 | | | | | | | | | x |
| *Ircinia* sp. 2 | | | | | x | | | | |
| *Jaspis* sp. | | x | x | | | | | | |
| *Monanchora arbuscula* [48] | xx | | | xxx | | | | xx | x |
| *Neopetrosia* sp. | | | | | x | | | | |
| *Niphates amorpha* [59] | | | | | | | | | x |
| *Oceanapia* sp. | | x | x | | xx | | | | |
| *Petrosia* sp. | | x | | | | | | | |
| *Scopalina ruetzleri* [60] | x | | | xx | | | | x | x |
| *Spirastrella hartmani* [61] | x | xx | x | xx | xxx | x | x | | |
| *Suberites* sp. | | | | | x | | | | |
| *Thorecta* sp. | | x | | | | | | | |
| *Topsentia ophiraphidites* [62] | x | x | | xx | x | | | x | xx |
| Verongida unidentified | | | | | x | | | | |
| *Xestospongia muta* [47] | x | | | | | | | | |
| *Xestospongia* sp. | | | | | x | | | | |
| CLASS CALCAREA | | | | | | | | | |
| *Arturia* cf. *alcatraziensis* [63] | | | | | | | | | x |
| *Ascandra* cf. *atlantica* [64] | | | | x | | | | | |
| *Ascandra* sp. | | x | | | xx | x | x | | x |
| Calcarea unidentified | | | | | x | | | | |
| *Clathrina aurea* [65] | x | | | | | | | x | x |
| *Clathrina* aff. *luteoculcitella* [66] | | x | | x | | | | | |
| *Clathrina delicata* [67] | | x | | | | | | | |
| *Clathrina insularis* [39] | | | | x | | | | | x |
| *Clathrina* sp. 1 | | | | | | | | | x |
| *Clathrina* sp. 2 | | | | x | | | | | x |
| *Clathrina* sp. 3 | | | | x | | | | | |
| *Clathrina* sp. 4 | | | | x | | | | | |
| *Clathrina* sp. 5 | | | | x | | | | | |
| *Clathrina* sp. 6 | | x | | | | | | | |
| *Janusya* sp. | | x | | | | | | | |
| *Neoernsta* sp. 1 | | x | | | | | | | |
| *Neoernsta* sp. 2 | | | | x | | | | | |
| *Neoernsta* sp. 3 | | | | | | | | | x |
| CLASS HOMOSCLEROMORPHA | | | | | | | | | |
| cf. *Aspiculortis* sp. | | x | | | | | | | |
| *Oscarella* cf. *zoranja* [68] | | xx | | x | xx | | | | |
| *Oscarella* sp. 1 | | x | xx | | xx | | | | |
| *Oscarella* sp. 2 | | x | x | | x | | | | |
| *Oscarella* sp. 3 | | | x | | | | | | |

**Table 1.** *Cont.*

| Species | Sapata Cave | | | Ilha do Meio Cave | | | | Pedras Secas Tunnel | |
| --- | --- | --- | --- | --- | --- | --- | --- | --- | --- |
| | Outside | Semi-Dark Zone | Transition Zone | Outside | Semi-Dark Zone | Transition Zone | Dark Zone | Outside | Semi-Dark Zone |
| cf. *Oscarella* sp. | | | x | | | | | | |
| *Plakina coerulea* [69] | | x | | | | | | | |
| *Plakina* sp. 1 | | x | x | | | | | | |
| *Plakina* sp. 2 | | x | | | | | | | |
| *Plakinastrella microspiculifera* [35] | | | | x | xxx | | | | xx |
| *Plakinastrella* sp. 1 | | | | | x | | | | |
| *Plakinastrella* sp. 2 | | | | | x | | | | |
| *Plakortis angulospiculatus* [70] | xx | | | xx | | | | x | xxx |
| *Plakortis insularis* [35] | | xxx | | x | xx | | | | |
| *Plakortis microrhabdifera* [35] | x | | | | | | | | |
| *Plakortis spinalis* [38] | | x | | | | | | | |
| *Plakortis* sp. | | | | | xx | x | | | |
| Number of species | 13 | 28 | 10 | 22 | 24 | 5 | 3 | 13 | 26 |

The semi-dark zone of Sapata Cave has a very rich sponge community, with at least 28 species (Figure 4E–M, Table 1). The most abundant sponges in this zone are *Plakortis insularis*, *Diplastrella megastellata*, *Oscarella* cf. *zoranja*, *Spirastrella hartmani*, and *Ectyoplasia ferox*. Calcareous sponges such as *Clathrina delicata* are patchily distributed, especially within crevices and slits (Figure 4E).

The transition zone is poorer in sponge species, with only 10 species recorded so far (Figure 4N–P, Table 1). The most common species there are *Gastrophanella cavernicola* and *Diplastrella megastellata*. Other common sponges in the transition zone are undescribed species of the genera *Jaspis* and *Oscarella*. Most species in this zone are cream, yellow, or whitish in color (Figure 4).

The dark zone of Sapata Cave and its sponge community remain unexplored.

*3.2. Ilha do Meio Cave*

3.2.1. Description of the Cave

Ilha do Meio Cave, also called "Caverna da Meia-Lua" (Portuguese for "Half-Moon Cave"; Figure 5), is located on the NW side of Ilha do Meio, a small tabular island near the NE end of the archipelago (Figure 1B,D). The rocky shore outside the cave is mostly sub-horizontal, with a weak declivity down to 15 m depth (Figure 6A). It is colonized mainly by green and brown algae (*Caulerpa* sp., *Canistrocarpus cervicornis*, and *Dictyota intestinalis*). Near the island, there are many ledges forming small sheltered cavities of approximately 1–2 m long and high, which are colonized by a rich benthic community, especially sponges (Figure 6B).

The cave entrance is roughly triangular, small, approximately 4 m high by 7 m wide, and at 5–9 m depths (Figures 5 and 6C). It opens into a relatively small room, approximately 10 m long by 7 m wide, characterized as a semi-dark zone. The bottom is a mix of sand and rock (Figure 6C). The walls of this zone harbor a rich benthic fauna, especially sponges (Figure 6D). There is a short transition zone from 10–15 m of the entrance, with reduced benthic cover but also dominated by sponges (Figure 6E). The dark zone is formed by a roughly cylindrical passage, approximately 60 m long by 2–3 m in diameter, which becomes larger in some places forming two rooms up to 5 m wide, at 30 and 60 m from the entrance (Figure 5). It is almost azoic, with mainly bare rock colonized by only a few sponges, rare bryozoans, and serpulid polychaetes (Figure 6F). Only the first half of the dark zone was explored in this study (Figure 5). The cave has a single entrance and water circulation inside is very weak.

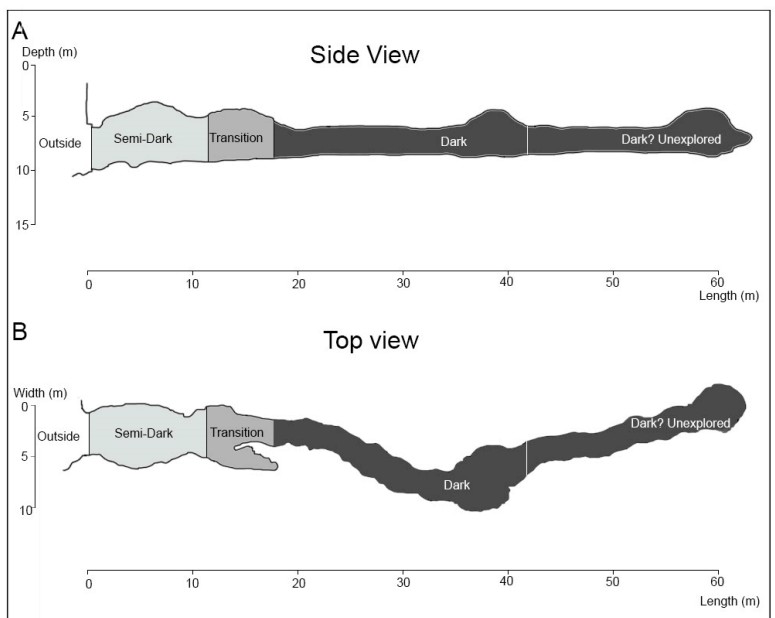

**Figure 5.** Schematic sections of Ilha do Meio Cave. (**A**) Lateral view. (**B**) Top view.

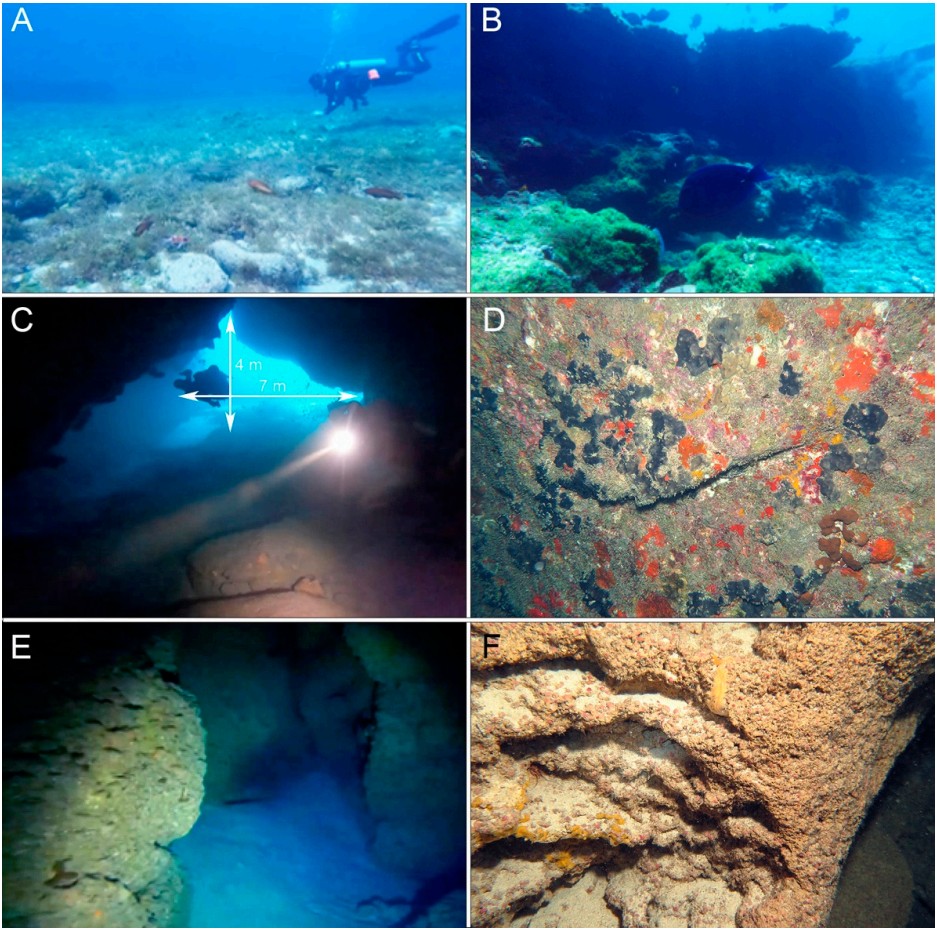

**Figure 6.** Ilha do Meio Cave. (**A**) Sub-horizontal rocky bottom outside the cave. (**B**) Ledges outside the cave. (**C**) Cave entrance and semi-dark zone viewed from inside the cave. (**D**) Benthic community in the semi-dark zone, dominated by the sponges *Plakinastrella microspiculifera* (dark grey) and *Spirastrella hartmani* (orange). (**E**) Transition zone. (**F**) Very poor benthic community in the dark zone, mostly with sparse encrusting demosponges (*Spirastrella hartmani*, *Diplastrella megastellata*).

The mobile fauna includes many crustaceans, such as the Banded Coral Shrimp *Stenopus hispidus* [71] and the Slipper Lobster *Scyllarides brasiliensis* [72], both of which live in crevices in the semi-dark zone of the cave. Red shrimps (cf. *Lysmata* sp.) live on the walls in the dark zone. The Atlantic Goliath Grouper *Epinephelus itajara* also finds shelter inside the cave, mainly in the semi-dark zone [46].

### 3.2.2. Sponge Communities of Ilha Do Meio Cave

The sponge community outside Ilha do Meio Cave is very rich and colorful, especially in the small shadowed cavities and overhangs (Figure 7A–C, Table 1). A total of 22 sponge species were recorded, of which the most common were *Agelas dispar*, *Monanchora arbuscula*, *Ircinia strobilina*, *Plakortis angulospiculatus,* and the calcareous *Clathrina* spp.

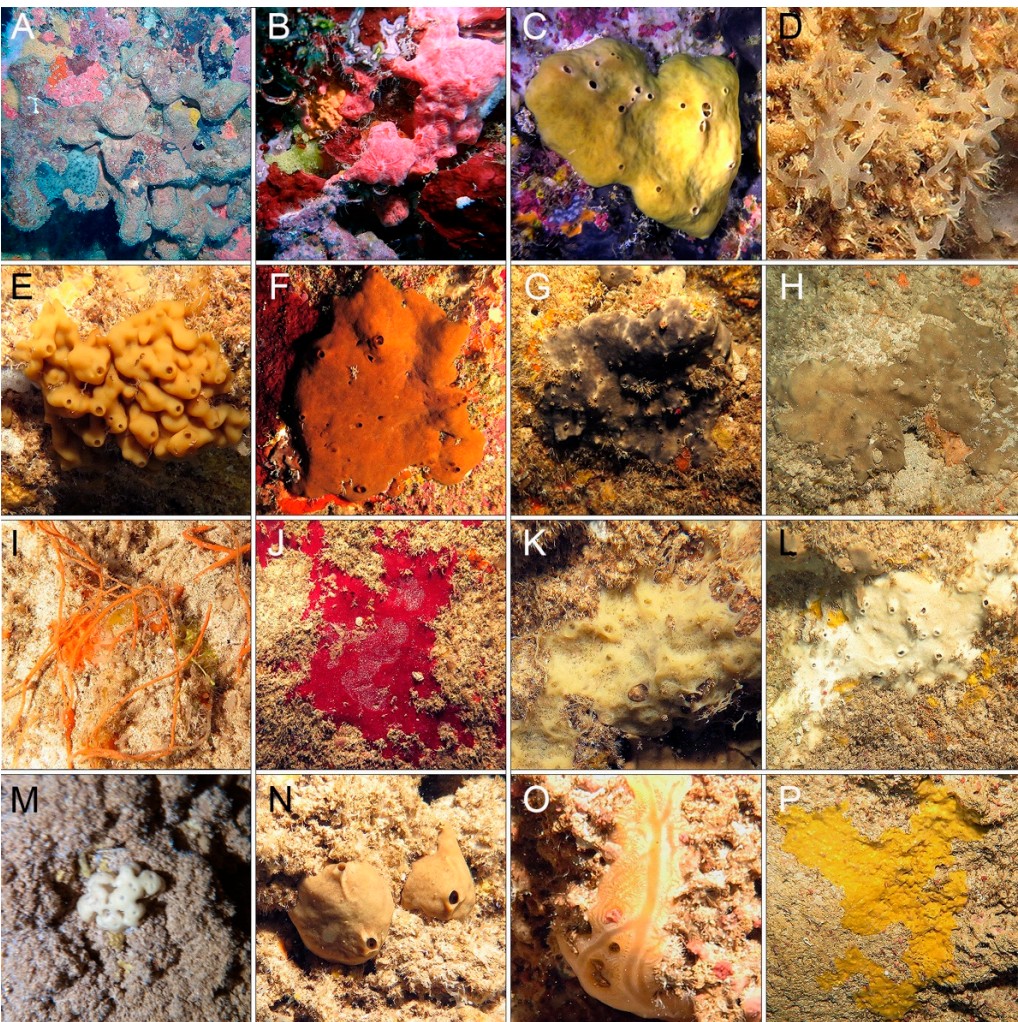

**Figure 7.** *In situ* photographs of representative sponge species living inside and outside Ilha do Meio Cave. (**A–C**) Outside the cave, in shadowy habitats under overhangs: (**A**) *Agelas dispar* (brown) and *Ircinia strobilina* (grey). (**B**) *Monanchora arbuscula* (pink) and *Clathrina insularis* (yellow). (**C**) *Plakortis angulospiculatus.* (**D–L**) Semi-dark zone: (**D**) *Ascandra* sp. (**E**) *Oscarella* cf. *zoranja.* (**F**) *Plakortis insularis.* (**G**) *Plakinastrella microspiculifera.* (**H**) *Plakinastrella* sp. (**I**) *Oceanapia* sp. (**J**) *Dercitus* (*Stoeba*) *latex.* (**K**) *Xestospongia* sp. (**L**) *Neopetrosia* sp. (**M,N**) Transition zone: (**M**) *Gastrophanella cavernicola.* (**N**) *Plakortis* sp. (**O,P**) Dark zone: (**O**) *Spirastrella hartmani.* (**P**) *Diplastrella megastellata.*

The sponge community in the semi-dark zone of Ilha do Meio Cave is very rich, with 24 species recorded (Figure 7D–L, Table 1), but less colorful than that outside the cave. The most abundant sponges in this zone are *Plakinastrella microspiculifera*, *Plakortis insularis*, *Spirastrella hartmani*, *Oscarella* cf. *zoranja*, and the calcareous *Ascandra* sp.

The transition zone of Ilha do Meio Cave is poor in sponges, with only five species recorded (Figure 7M–N, Table 1). Similar to the same zone in Sapata Cave, the dominant species are *Gastrophanella cavernicola* and *Diplastrella megastellata*. Most species in this zone are cream, yellow, light brown, or whitish in color.

The dark zone of Ilha do Meio Cave is very poor, with mostly bare rock and very few benthic organisms. Only three sponge species were found in this zone, all of them quite rare (Figure 7O,P, Table 1): *Diplastrella megastellata*, *Spirastrella hartmani*, and *Ascandra* sp. They are yellow, cream, or white in color.

### 3.3. Pedras Secas Tunnel
### 3.3.1. Description of the Tunnel

Pedras Secas ("Dry Rocks" in Portuguese) is a rocky reef located 1.5 km east of the main island, on its exposed side ("Mar de Fora"; Figure 1B,D). It has a rugged relief, with many overhangs and tunnels colonized by a rich benthic fauna dominated by algae and sponges. Pedras Secas Tunnel is the largest underwater tunnel at the "Pedras Secas 1" diving site. It is roughly cylindrical, straight, and approximately 15 m long by 3 m in diameter (Figures 8 and 9). Both sides open at 15–18 m depths. The bottom is mostly sandy, with few rocks. The water circulation is very high, with strong waves and currents being funneled within the tunnel. The illumination is relatively high, comparable to the semi-dark zones near the entrance of the Sapata and Ilha do Meio caves.

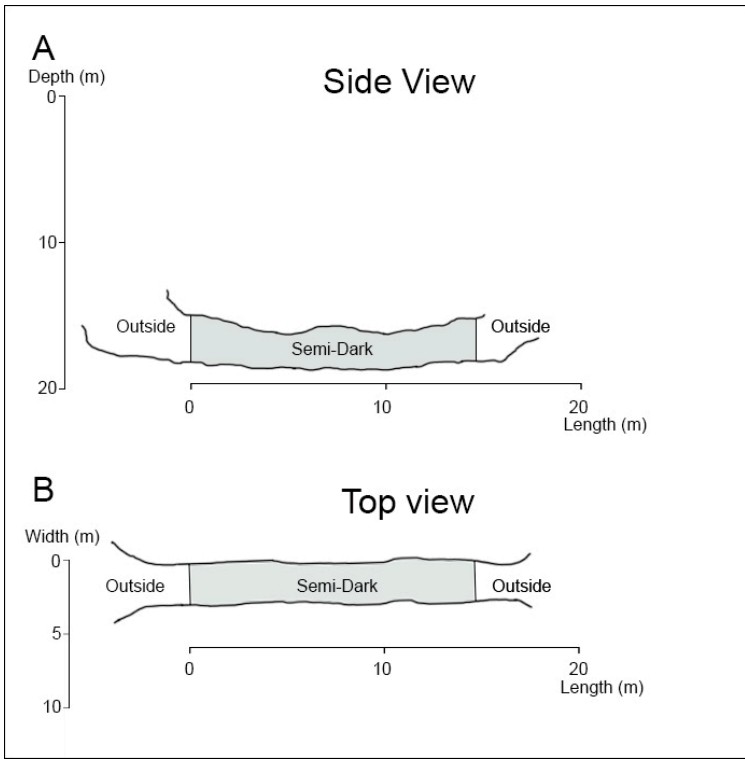

**Figure 8.** Schematic sections of Pedras Secas Tunnel. (**A**) Lateral view. (**B**) Top view.

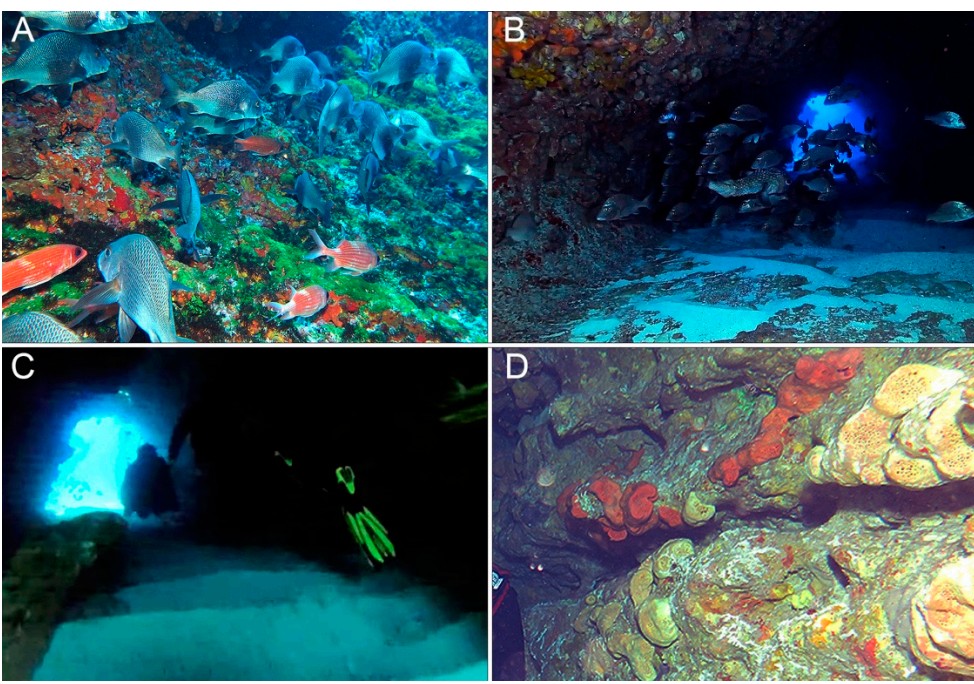

**Figure 9.** Pedras Secas Tunnel. (**A**) Rocky bottom outside the tunnel with abundant fish, green algae, and sponges. (**B**,**C**) General views of the tunnel. (**D**) Wall in the semi-dark zone at the middle of the tunnel, dominated by the sponges *Plakortis angulospiculatus* (brown), *Agelas dispar* (light brown), and *Agelas clathrodes* (orange).

### 3.3.2. Sponge Communities of Pedras Secas Tunnel

Outside the tunnel, Pedras Secas has a rich and colorful sponge community. We recorded 13 sponge species, but there are probably more to be discovered. The most common sponges are *Aiolochroia crassa*, *Agelas dispar*, *Ircinia strobilina*, *I. felix*, *Clathria minuta*, and *Monanchora arbuscula* (Figure 10A,B, Table 1).

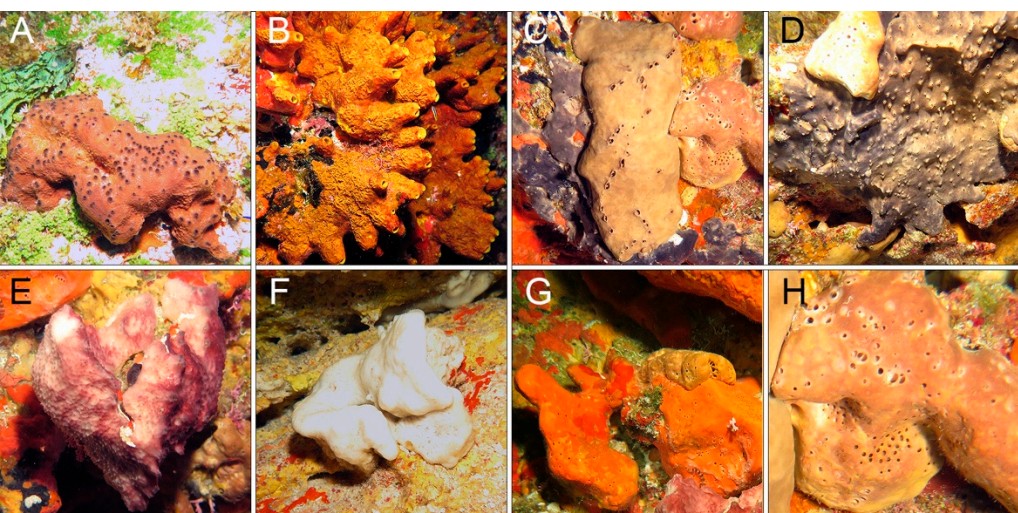

**Figure 10.** *In situ* photographs of representative sponge species living inside and outside Pedras Secas Tunnel. (**A**,**B**) Outside the tunnel: (**A**) *Ircinia felix*. (**B**) *Aiolochroia crassa*. (**C–H**) Semi-dark zone: (**C**) *Plakortis angulospiculatus*. (**D**) *Plakinastrella microspiculifera*. (**E**) *Ircinia* sp. (**F**) *Topsentia ophiraphidites*. (**G**) *Agelas clathrodes*. (**H**) *Agelas dispar*.

The semi-dark zone inside the tunnel has a very rich sponge community, with at least 26 species (Figure 10C–H, Table 1). The most common sponges there are *Plakortis angulospiculatus*, *Agelas dispar*, *A. clathrodes*, *Plakinastrella microspiculifera,* and *Topsentia ophirhaphidites*.

### 3.4. Species Richness and Composition

In the three submarine cavities studied, the semi-dark zone is the richest zone in terms of the number of sponge species, with 24 to 28 species (Figure 11). In Sapata Cave and Pedras Secas Tunnel, the semi-dark zones are twice as rich as outside the cavities. In Ilha do Meio, the rocky coast outside the cave is richer in sponges than in the other two sites (22 species, against 13 spp. in Sapata Cave and in Pedras Secas Tunnel), mainly due to the presence of many small shadowy cavities and ledges near the cave entrance (Figure 6B). The transition zones in the two caves (5–10 spp.) and especially the dark zone in Ilha do Meio Cave (3 spp.) are much poorer in sponges than the semi-dark zones (Figure 11).

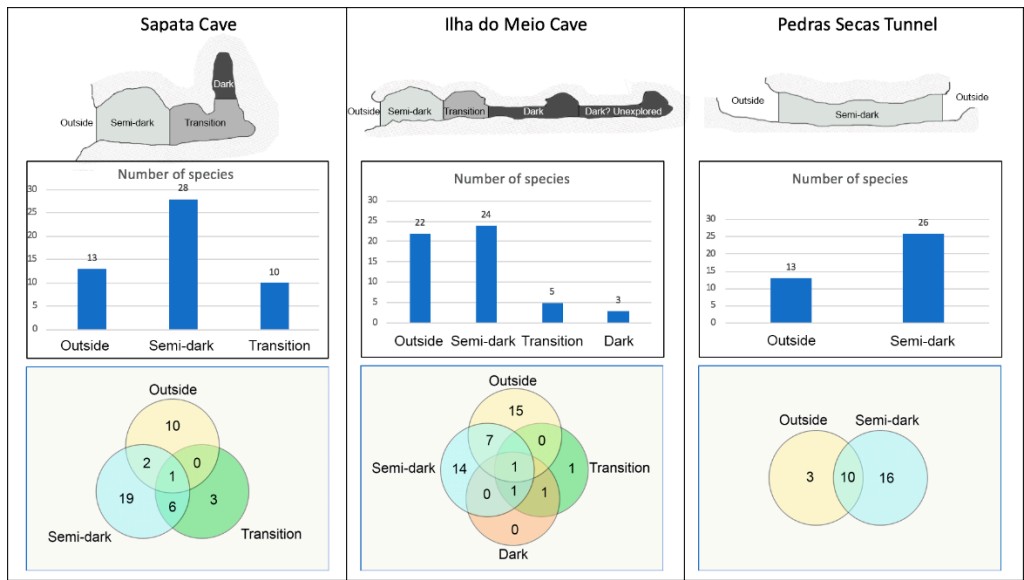

**Figure 11.** Schematic drawings in lateral views, species richness (bar charts), and number of exclusive and shared species (Venn diagrams) in different zones inside and outside the caves and tunnel studied on Fernando de Noronha.

In the Sapata and Ilha do Meio caves, most species are exclusive to each zone, with very few species shared between different zones. In contrast, the semi-dark zone of Pedras Secas Tunnel shares 10 out of 16 species with the rocky reef outside the cave (Figure 11).

In all three cavities, Demospongiae was the most diverse sponge class, with 47% to 65% of the species, followed by Homoscleromorpha in the two caves (31% to 34%) and Calcarea in Pedras Secas Tunnel (27%; Figure 12). Demospongiae was also the most diverse class in each individual zone of the three cavities (Figure 12).

With all sponge classes summed up, the total number of species was greater in Sapata Cave than in Ilha do Meio Cave and Pedras Secas Tunnel (Figure 13A). The number of shared species was greater between the two caves (8 spp.) than between each cave and the tunnel (3–4 spp.). Only two species (*Topsentia ophiraphidites* and *Ascandra* sp.) were shared by the three cavities studied (Figure 13B, Table 1).

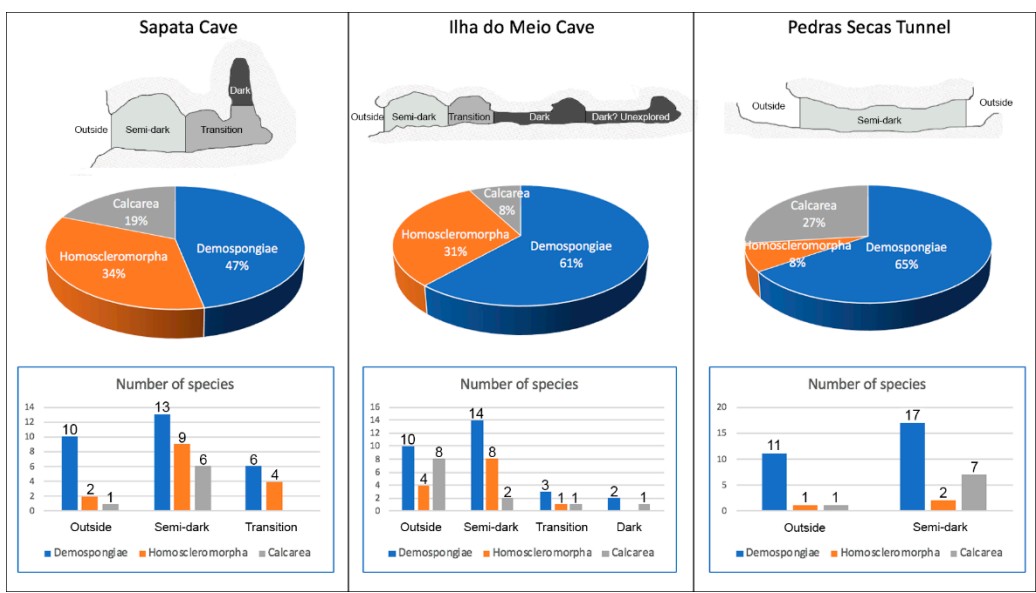

**Figure 12.** Schematic drawings in lateral views, proportion of species of each sponge class inside the caves and the tunnel studied (pie charts—all zones summed up), and number of species of each class in different zones inside and outside the cavities (bar charts).

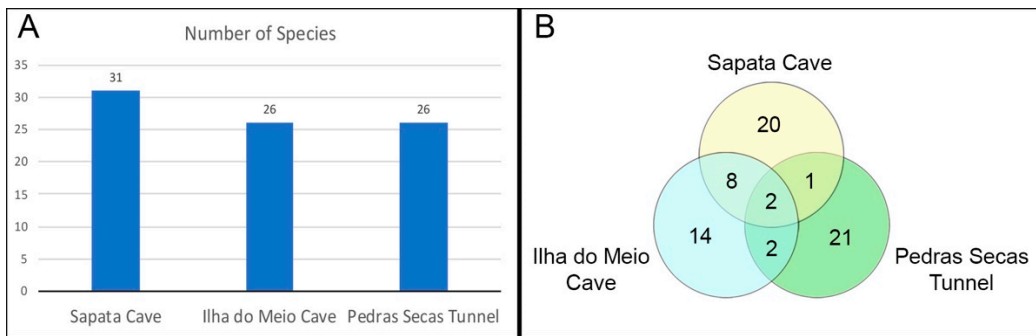

**Figure 13.** Comparison of the whole sponge communities living inside the three submarine cavities studied on Fernando de Noronha. (**A**) Species richness. (**B**) Venn diagram.

## 4. Discussion

Overall, our results show that the submarine caves and tunnels of the Fernando de Noronha Archipelago harbor very rich and diverse sponge communities, with 69 species compared to 29 spp. found outside the cavities in our study. The sponge diversity in the Fernando de Noronha caves is similar to that of Mediterranean caves, where 329 species were reported from 185 caves and a maximum of 86 species from a single cave [6]. The high diversity of sponges in submarine caves may be explained by the absence of competition with photophilic species (especially algae), habitat complexity, and representation of deep-sea habitats [6,73,74].

This study also shows some interesting differences between single-entrance caves and tunnels. The two caves have strong physical and biological gradients from the exterior to the interior, with low light, low water circulation, and probably low food. Sponge diversity and abundance are higher in the semi-dark zone near the cave entrance, reducing in darker zones. These gradients in community structure are probably related to the well-known dramatic physical gradients of light and water circulation from the exterior to the interior of caves, but possibly also to differences in recruitment, food availability, and spatial competition in different zones of the caves [6,75,76]. There is also a reduction in colors towards the interior of the caves: outside the caves, sponges are very colorful, including brown, black, gray, green, bright yellow, purple, red, orange, pink, cream, and white colors;

in the semi-dark zone, most species are brown, gray, red, orange, cream, or white; and in the transition and dark zones, cream-colored, light yellow, and white species predominate. This gradient may be explained by the UV protection given by black (melanin), red, orange, or bright yellow pigments (carotenoids), or to an aposematic role of bright colors against predators, both of which would be useless in dark environments. It could also be related to the influence of green, blue, and purple microalgae and cyanobacteria, which are often associated with sponges in light-exposed environments (discussed by, e.g., [77,78]). The reduction in bright colors inside caves goes against the hypothesis that color is a random, non-adaptive result of some metabolic product of sponges.

In the two caves, there are greater changes in species composition and more exclusive cave species than in the tunnel. Due to its short length, Pedras Secas Tunnel has no transition or dark zone. The differences between its semi-dark zone and the outside are smaller than in the caves, with weaker physical and biological gradients: there is more light, higher water circulation, and probably more food available for filter feeders. Sponges are diverse and abundant throughout the tunnel, with more colorful and more common reef species than in the caves.

The general patterns of sponge abundance, diversity, and zonation of the caves on Fernando de Noronha are very similar to those found in Mediterranean caves [6,15,73]. In both regions, the richest sponge communities in terms of abundance and species diversity are found in the semi-dark zone near the cave entrance. This zone also shows the highest diversity of morphological sponge types, especially in the Eastern Mediterranean [15]. The dark zones inside Atlantic and Mediterranean marine caves are usually very poor in benthic species, often mostly with bare rock [6]. This study is the first to report a gradient of external sponge colors, with a reduction in colorful species from the exterior of the caves towards the dark zones deep inside the caves, although it can also be noted in some Mediterranean caves (see Figure 21 in [6]). The underwater caves on Fernando de Noronha harbor interesting relict, deep sea, and poorly studied taxa such as desma-bearing sponges ("lithistids") and the class Homoscleromorpha, similar to other regions such as the Northwestern Atlantic Ocean, the Mediterranean Sea, and the Pacific Ocean [6,15–18,79].

The spongiofauna found inside the caves is usually different from outside, but that may depend on the geomorphology of the area. At the Sapata and Ilha do Meio caves, respectively, only three and eight species were shared between outside and inside the caves. These results support the hypothesis that the species living within marine caves are highly specialized and typical of these cryptic marine ecosystems. Although deep-sea species are often found in shallow submarine caves [10,12,13,79] and a desma-bearing sponge was abundant in the Sapata and Ilha do Meio caves (*Gastrophanella cavernicola*), none of the species found in the Fernando de Noronha caves have been reported from deeper areas so far. The limited availability of light, nutrients, and water circulation in completely closed underwater caves plays an important role in shaping the sponge communities' composition, diversity, and biology [6]. This limitation may be mitigated at sites where these abiotic factors are more similar to the water outside the underwater cave. For instance, at Pedras Secas Tunnel, where there are two entrances and, therefore, the water current is stronger, there is a higher similarity when comparing inside with outside, with 10 out of 16 species shared with the rocky reef. The higher water movement may also explain the higher percentage of calcareous sponges in this tunnel when compared with the two caves.

This study has allowed a great increase in the number of cave-dwelling sponge species known to dwell in caves and tunnels of the Fernando de Noronha Archipelago. Previous works [33–35,37] reported 29 species from the same three underwater cavities studied here, with eight species in Sapata Cave, four species in Ilha do Meio Cave, and 20 species in Pedras Secas Tunnel. We increased these numbers to 32 species in Sapata Cave, 26 species in Ilha do Meio Cave, and 26 species in Pedras Secas Tunnel, with a total of 69 cave-dwelling species (an increase of 138%; Table 1; Supplementary Table S1). Furthermore, there are still several other species to be discovered in these cavities, demonstrating that the sponge diversity in caves and tunnels of Fernando de Noronha was greatly underestimated and

needs to be better studied. The low number of species shared by the three cavities (only two) indicates that each cavity has a unique composition of sponge species, and, therefore, the study of other caves and tunnels in the area might result in a great increase in known diversity. The uniqueness of cave communities also has important implications for cave conservation and management, since it shows that the protection of many caves is necessary if it is to cover a proper representation of the species and ecological processes taking place in different caves [6,75].

This is the first study describing sponge communities in submarine caves and tunnels in Brazil. The study of submarine cavities in Brazil is still incipient, and many additional steps are needed for a better knowledge of marine caves and tunnels along the Brazilian coast and oceanic islands: (1) to provide quantitative data and detailed mappings using photoquadrats, photogrammetry, transects, echo-sounders, light sensors, and compasses [80,81]; (2) to investigate the dark zones of Sapata Cave and Ilha do Meio Cave, which remain largely unexplored; (3) to carry out more collections of sponges in the three cavities studied here, because many species appeared only in photographs and were not collected yet; (4) to publish taxonomic descriptions of the species that remain undescribed; (5) to investigate the planktonic and sponge-associated microbiota in caves and tunnels on Fernando de Noronha; and (6) to explore other caves and tunnels on Fernando de Noronha and in other localities in Brazil such as Abrolhos, the Rocas Atoll, Trindade Island, Rio de Janeiro, and Arraial do Cabo.

**Supplementary Materials:** The following supporting information can be downloaded at: https://www.mdpi.com/article/10.3390/jmse12040657/s1, Table S1: List of sponge species found inside and outside the caves and tunnel studied in Fernando de Noronha, with species authorities and Aphia IDs from WoRMS.

**Author Contributions:** Conceptualization, G.M.; data curation, C.B.P.; formal analysis, G.M.; funding acquisition, G.M., M.K., U.P. and M.V.L.; investigation, G.M., A.L., J.S., U.P., C.B.P. and M.V.L.; methodology, G.M.; project administration, G.M., M.K., U.P. and M.S.L.; supervision, G.M.; writing—original draft, G.M.; writing—review and editing, A.L., J.S., M.K., U.P., M.S.L., B.F.R.d.O. and M.V.L. All authors have read and agreed to the published version of the manuscript.

**Funding:** G.M. received grants from Conselho Nacional de Desenvolvimento Científico e Tecnológico (CNPq-31/2019), grant number 443302/2019-7, and Fundação Carlos Chagas Filho de Amparo à Pesquisa do Estado do Rio de Janeiro (FAPERJ CNE 2018), grant number E-26/202.898/2018-BOLSA. C.B.P. received a fellowship from Conselho Nacional de Desenvolvimento Científico e Tecnológico (CNPq-PROTAX 22/2020), grant number 441747/2020-5. A.L. and J.S. received post-doctoral fellowships from Fundação Carlos Chagas Filho de Amparo à Pesquisa do Estado do Rio de Janeiro (grants FAPERJ Pós-Doutorado Nota 10 numbers E-26/201.977/2020 and E-26/204.366/2021, respectively). M.S.L received grants from FAPERJ (grant numbers E-26/211.284/2021 and E-26/200.948/2021) and CNPq (grant number 309158/2023-0). M.V.L received a scholarship from Fundação Coordenação de Aperfeiçoamento de Pessoal de Nível Superior (CAPES; grant number 88887.815215/2023-00). U.P. received a Research Productivity fellowship from CNPq (grant number CNPq 310914/2021-3). M.K. received grants from Fundação Carlos Chagas Filho de Amparo à Pesquisa do Estado do Rio de Janeiro (FAPERJ grant numbers SEI-260003/001170/2020 and E-26/200.912/2021) and a fellowship from Conselho Nacional de Desenvolvimento Científico e Tecnológico (CNPq grant number 306977/2021-4).

**Institutional Review Board Statement:** Not applicable.

**Informed Consent Statement:** Not applicable.

**Data Availability Statement:** Collection and identification data for the specimens collected in this study are partially available at http://labpor.hopto.org/fmi/webd/Porifera_MNRJ-30-04-2023, accessed on 6 March 2020.

**Acknowledgments:** The authors are grateful to the Instituto Chico Mendes de Conservação da Biodiversidade (ICMBio) for the license to collect on Fernando de Noronha (SISBIO license number 73826-2). We also thank Atlantis Divers for its diving support; Fernando Moraes and Zaira Matheus for information on the geomorphology of the caves and tunnel studied; and Pousada Água Viva for

support during the expeditions to Fernando de Noronha. We are also grateful to Mariana Luiz for her help with the identification of calcareous sponges.

**Conflicts of Interest:** The authors declare no conflicts of interest. The funders had no role in the design of this study; in the collection, analyses, or interpretation of data; in the writing of the manuscript; or in the decision to publish the results.

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
