# Peer review of "Sponge Communities of Submarine Caves and Tunnels on the Fernando de Noronha Archipelago, Northeast Brazil"

_jmse, doi:10.3390/jmse12040657_

Round 1

Reviewer 1 Report

Comments and Suggestions for Authors

This is a clear and well-written manuscript about sponge communities of submarine caves and tunnels in Brazil. The manuscript presents a long species list (69 taxa), semi-quantitative abundance data, as well as several photographs and maps illustrating sponge diversity and community patterns in caves of the study area. Information about sponges in Brazilian marine caves and caves of the Southern Atlantic is very limited; therefore, this manuscript fills important gaps and increases our understanding of sponge community patterns in Atlanto-Mediterranean marine caves. In addition, the authors report on an interesting gradient of sponge external colours towards the dark zones inside the caves which has not been properly highlighted in previous studies. The manuscript also fits well with the topic of the special issue “5th International Workshop on Taxonomy of Atlanto-Mediterranean Deep-Sea & Cave Sponges”. I support publication after minor revisions (see comments below):

Line 106: change “varying from 24–30°C” to “varying from 24 to 30 °C.

Line 133: change “We made two expeditions” to “Two sampling expeditions were organized” or similar.

Line 157: change “ranging from 7–17 m depth” to “ranging from 7 to 17 m depth” or “at a depth range 7–17 m”

Line 159: delete “with”

Lines 160-161: modify text to “a transition zone, at a distance of 15–30 m from the entrance, with low light”

Lines 197-198: modify text to “Other common sponges in the transition zone are undescribed species of the genera Jaspis and Oscarella.”

239: delete “with”

Lines 247-247: Where there any other sessile animals such as serpulid polychaetes or bryozoans?

Line 286: replace “shadowed” with “shadowy”

Line 300: delete “with”

Line 338: replace “shadowed” with “shadowy”

Line 353: Secal or Secas?

Line 402: Also see https://doi.org/10.3389/fmars.2021.630900

Lines 406-410: Is it possible that some of the species found inside caves are also found in mesophotic or deeper waters of the broader area?

Lines 411-414: Could the higher percentage of Calcarea (line 353 and Figure 12) in this tunnel be related to the higher hydrodynamism?

Lines 430-432: Rephrase to “to provide quantitative data and detailed mappings using photoquadrats, photogrammetry, transects, echo-sounders, light sensors, and compasses”

Table 1: Since Table 1 does not provide authorities, I suggest adding a supplementary file sponges per cave/tunnel, including authorities and AphiaID (from WoRMS) at least for those sponges which were identified at species level. In addition, cf. is italicized for cf. Aspiculortis sp. while it was not written in italics for other species (e.g. Arturia cf. alcatraziensis).  

Figure 8. If the zones named “Outside” have a ceiling then they could be named as “Entrance”

Figures 11 and 12: Please remove the red lines below the names of the cavities (Sapata, Meio, Pedras Secas). The word “Semi-Dark” is not hyphenated in the main text. Write it in the same way throughout the manuscript.

Author Response

Line 106: change “varying from 24–30°C” to “varying from 24 to 30 °C.

- OK, done.

Line 133: change “We made two expeditions” to “Two sampling expeditions were organized” or similar.

- OK, done.

Line 157: change “ranging from 7–17 m depth” to “ranging from 7 to 17 m depth” or “at a depth range 7–17 m”

- OK, done.

Line 159: delete “with”

- OK, done.

Lines 160-161: modify text to “a transition zone, at a distance of 15–30 m from the entrance, with low light”

- OK, done.

Lines 197-198: modify text to “Other common sponges in the transition zone are undescribed species of the genera Jaspis and Oscarella.”

- OK, done.

Line 239: delete “with”

- OK, done.

Lines 247-247: Where there any other sessile animals such as serpulid polychaetes or bryozoans?

- OK, we added “rare bryozoans and serpulid polychaetes” after “a few sponges” (line 248)

Line 286: replace “shadowed” with “shadowy”

- OK (now line 289)

Line 300: delete “with”

- OK (now in line 303)

Line 338: replace “shadowed” with “shadowy”

- OK (now in line 342)

Line 353: Secal or Secas?

- OK, corrected to “Pedras Secas” (now in line 358)

Line 402: Also see https://doi.org/10.3389/fmars.2021.630900

- OK, we added a new reference [93]: Pisera & Gerovasileiou, 2021 (now in line 403, complete reference listed in lines 708–709). Former references [88] and [89] became [94] and [95], respectively.

Lines 406-410: Is it possible that some of the species found inside caves are also found in mesophotic or deeper waters of the broader area?

-OK. We added a new sentence in lines 429–433: “Although deep-sea species are often found in shallow submarine caves [11,13,14,90] and a desma-bearing sponge was abundant in Sapata and Ilha do Meio Caves (Gastrophanella cavernicola), none of the species found in Fernando de Noronha caves has been reported from deeper areas so far”.

Lines 411-414: Could the higher percentage of Calcarea (line 353 and Figure 12) in this tunnel be related to the higher hydrodynamism?

- Yes. We added a sentence in lines 439–441: “The higher hydrodynamism may also explain the higher percentage of calcareous sponges in this tunnel when compared to the two caves.”

Lines 430-432: Rephrase to “to provide quantitative data and detailed mappings using photoquadrats, photogrammetry, transects, echo-sounders, light sensors, and compasses”

- OK. Done as suggested (now in lines 461–463)

Table 1: Since Table 1 does not provide authorities, I suggest adding a supplementary file sponges per cave/tunnel, including authorities and AphiaID (from WoRMS) at least for those sponges which were identified at species level. In addition, cf. is italicized for cf. Aspiculortis sp. while it was not written in italics for other species (e.g. Arturia cf. alcatraziensis).  

- OK. We added a Supplementary Table 1 with all the species authorities and Aphia IDs from WoRMS (Supplementary table cited in line 448).

- In addition, “cf.” is now italicized in cf. Aspiculortis sp.

Figure 8. If the zones named “Outside” have a ceiling then they could be named as “Entrance”

- The zones named “Outside” do not have a ceiling, so the name of these zones was left unchanged.

Figures 11 and 12: Please remove the red lines below the names of the cavities (Sapata, Meio, Pedras Secas). The word “Semi-Dark” is not hyphenated in the main text. Write it in the same way throughout the manuscript.

- OK. The red underlines were removed in figs. 11 and 12. However, the word “Semi-dark” was already hyphenated in the main text, so it was left unchanged here and throughout the MS.

Reviewer 2 Report

Comments and Suggestions for Authors

This is an interesting and well-written study describing the submarine cave systems and their associated sponge communities in the Fernando de Noronha Archipelago, Brazil. The authors provide the first detailed characterization of the geomorphology and sponge fauna of three important caves/tunnels in this region. The work significantly expands the current knowledge of cave-dwelling sponges in the southwestern Atlantic Ocean, an area that has been poorly studied compared to other regions like the Mediterranean or the Caribbean. Overall, the study is valuable and relevant, and I recommend it for publication in Journal of Marine Science and Engineering after addressing the following comments:

At line 32, correct “environments”

Uniform the white balance across the photos. In many photos, the colour temperature is too low, and they look too yellow.

In the discussion, the authors could discuss the potential implications of the findings for cave conservation and management, given the high diversity and unique nature of these sponge communities.

Comments on the Quality of English Language

English is good. Some words are a bit archaic, but overall the article is easy to read

Author Response

At line 32, correct “environments”

- OK, done.

Uniform the white balance across the photos. In many photos, the colour temperature is too low, and they look too yellow.

- OK. The White balance was adjusted in figures 1, 3, 4, 6, 7, 9, and 10.

In the discussion, the authors could discuss the potential implications of the findings for cave conservation and management, given the high diversity and unique nature of these sponge communities.

-OK. A new sentence was added in lines 454-457 to explain this issue: “The uniqueness of cave communities also has important implications for cave conservation and management, since it shows that the protection of many caves is necessary to include proper representation of the species and ecological processes taking place in different caves [6,89].

Reviewer 3 Report

Comments and Suggestions for Authors

the manuscript "Sponge communities of submarine caves and tunnels in Fernando de Noronha Archipelago, Northeast Brazil" of Muricy et al, proposed for the publication is a very clear and symple product. The simplicity (absence of enhanced plans for studying distribution and/or correlation of results with abiotic features) has to be cvonsidered, however, as a limit in any of the proposed discussion. Authors do not know the actual limits of environmental situations inside the cave and this point, as others, could be proposed for the future (= not general calling to deepest studies, but what the Authors would desire to know better, according to their experience in the same caves).

but the discussion could be enriched also (and symply) by checking what other researchers suggested in their papers.

my compliments to images, and rich illustration of the manuscript.

suggestion and requests are directly indicated on the pdf file here annexed

Author Response

JMSE 2950335

Cover Letter - Response to the reviewers

Dear Editors of JMSE, special volume on the 5th International Workshop on Taxonomy of Atlanto-Mediterranean Deep-Sea & Cave Sponges,

Thank you for the revision of our manuscript. The numbers of the references in the MS are correct, but we forgot to update their numbers in the replies to the reviewers. Please find below our response to reviewer #3 with the correct reference numbers. All changes are highlighted in yellow.

We hope that the MS is now acceptable for publication in JMSE and we look forward for an editorial decision. Yours sincerely,

Guilherme Muricy

---------------------

Line 47: to replace “considered” by “discovered as” hot spots

- OK, done.

Line 54: to replace “Few organisms can survive in the inhospitable” by “many organisms can adapt to survive in the” environments of submarine caves.

- OK. Replaced by “Not many organisms can adapt to survive in the...”

Line 160: to replace “from 15–30 m of” by “at 15–30 m from”

- OK. Replaced by “at a distance of 15–30 m from”, as suggested by reviewer #1.

Line 164: To replace “low” by “weak”.

- OK, done.

Line 374: “...harbor very rich and diverse sponge communities”. Here the comparisons with other situations should be mentioned to justify such a statement.

- OK. We added a sentence to explain this issue: ...”, with 69 species, compared to 29 spp. found outside the cavites in our study. Sponge diversity in Fernando de Noronha caves is similar to that of Mediterranean caves, with 329 species reported from 185 caves and a maximum of 86 species from a single cave [6].” (lines 378–381).

Lines 377–378: “Sponge diversity and abundance are higher in the semi-dark zone near the cave entrance, reducing in darker zones.” Please check Busasaotti et al., 2006; Denitto et al., 2007 for a discussion/interpretation of species richenss gradients in submarine caves.

- OK. We added a new sentence to explain this issue: “These gradients in community structure are probably related to the well-known dramatic physical gradients of light and water circulation from the exterior to the interior of caves, but possibly also to differences in recruitment, food availability and spatial competition in different zones of the caves [6,75,76].” (now in lines 389–392).

Lines 382–383: “in the transition and dark zones cream-colored, yellow and white species predominate.” What colors should represent? Why such an interest on colors? do you have indications from literature? have you an opinon? Please insert such elements of discussions.

-OK. We added two new sentences in lines 397–404 to clarify this issue: “This gradient may be explained by the UV protection given by black (melanin), red, orange or bright yellow pigments (carotenoids), to an aposematic role of bright colors against predators, both of which would be useless in dark enviroments. It could also be related to the influence of green, blue and purple microalgae and cyanobacteria, which are often associated to sponges in light-exposed environments (discussed by, e.g., [77,78]). The reduction of bright colors inside caves calls against the hypothesis that color is a random, non adaptative result of some metabolic product of sponges.”

Lines 396–397: “This study is the first to report a gradient of sponge external colors”. ... but without any discussion – explanation.

- OK – See previous comment; we also added a new sentence here, to indicate that this a more common phenomenon: “although it can also be noted in Mediterranean caves (see figure 21 in [6]).”  (now in lines 420–421)

Line 407: To replace “exclusive” by “typical”.

- OK, done. (now in line 430)

Lines 425–426: “therefore the study of other caves and tunnels in the area might result in a great increase in its known diversity.” Please refer to Bianchi and Gerovasilieu 2022 or Onorato and Belmonte 2018 for discussion elements on the high species richness of caves.

- OK. These references are now cited in a new sentence, added in lines 381–384: “The high diversity of sponges in submarine caves may be explained by the absence of competition with photophilic species (especially algae), habitat complexity and representation of deep-sea habitats [6,73,74].

Guilherme Muricy

April 11, 2024

Round 2

Reviewer 3 Report

Comments and Suggestions for Authors

approval of modifications inserted in the ms.